# Carbon Nanodots Attenuate Lipid Peroxidation in the LDL Receptor Knockout Mouse Brain

**DOI:** 10.3390/antiox12051081

**Published:** 2023-05-11

**Authors:** Keith M. Erikson, Kristina El-Khouri, Radmila Petric, Chenhao Tang, Jinlan Chen, Delicia Esther Cardenas Vasquez, Steve C. Fordahl, Zhenquan Jia

**Affiliations:** 1Department of Nutrition, University of North Carolina, Greensboro, NC 27401, USA; 2Department of Biology, University of North Carolina, Greensboro, NC 27401, USA; 3Institute for the Environment, University of North Carolina, Chapel-Hill, NC 27517, USA

**Keywords:** brain, lipid peroxidation, carbon nanodots, iron, copper, midbrain

## Abstract

Abnormal cholesterol metabolism can lead to oxidative stress in the brain. Low-density lipoprotein receptor (LDLr) knockout mice are models for studying altered cholesterol metabolism and oxidative stress onset in the brain. Carbon nanodots are a new class of carbon nanomaterials that possess antioxidant properties. The goal of our study was to evaluate the effectiveness of carbon nanodots in preventing brain lipid peroxidation. LDLr knockout mice and wild-type C57BL/6J mice were treated with saline or 2.5 mg/kg bw of carbon nanodots for a 16-week period. Brains were removed and dissected into the cortex, midbrain, and striatum. We measured lipid peroxidation in the mouse brain tissues using the Thiobarbituric Acid Reactive Substances Assay and iron and copper concentrations using Graphite Furnace Atomic Absorption Spectroscopy. We focused on iron and copper due to their association with oxidative stress. Iron concentrations were significantly elevated in the midbrain and striatum of the LDLr knockout mice compared to the C57BL/6J mice, whereas lipid peroxidation was greatest in the midbrain and cortex of the LDLr knockout mice. Treatment with carbon nanodots in the LDLr knockout mice attenuated both the rise in iron and lipid peroxidation, but they had no negative effect in the C57BL/6J mice, indicating the anti-oxidative stress properties of carbon nanodots. We also assessed locomotor and anxiety-like behaviors as functional indicators of lipid peroxidation and found that treatment with carbon nanodots prevented the anxiety-like behaviors displayed by the LDLr knockout mice. Overall, our results show that carbon nanodots are safe and may be an effective nanomaterial for combating the harmful effects caused by lipid peroxidation.

## 1. Introduction

Atherosclerosis can significantly increase the risk for coronary heart disease, stroke, and neurodegenerative diseases (Alzheimer’s disease) [1,2]. Studies have shown that inflammation plays a central role in the occurrence and development of atherosclerosis. Low-density lipoprotein receptor (LdLr) knockout mice are one of the most commonly used mouse models and have been widely used to study atherosclerosis and hyperlipidemia [3]. LdLr can bind to LDL particles to clear LDL from circulation. LdLr knockout mice have higher levels of LDL as well as inflammation biomarkers, including cytokines such as TNF-alpha and IL-1beta, sharing symptoms with those found in humans with dyslipidemia [3]. The atherosclerotic process in LdLr knockout mice is accelerated when the animals are fed an atherogenic diet, leading to atherosclerotic lesions and inflammation, which are usually observed after 16 weeks on the atherogenic diet [4].

Copper and iron are essential trace metals necessary for the proper functioning of the human body and must remain in the optimal physiological range for health [5]. These metals play important roles in cellular responses, signaling, regulation, and overall cell function and metabolism, making them necessary for normal brain function [5]. Inflammation is present in many common diseases, and systemic inflammation has been linked to alterations in trace metal homeostasis [6]. When the management of these metals becomes altered or disrupted, neurodegenerative processes can ensue, leading to the onset of neurodegenerative diseases (Alzheimer’s disease and Parkinson’s disease) [6]. When iron accumulates in the brain, the mitochondrial respiratory chain becomes impaired, which can lead to a large increase in levels of reactive oxygen species, which can damage proteins and result in cell death [7]. There are mechanisms in place to help manage iron homeostasis; however, during chronic inflammation, these mechanisms can be overwhelmed, leading to abnormal iron uptake at the blood-brain barrier [7]. Chronic inflammation has been shown to limit iron availability, while tissue-specific brain iron toxicity can occur without an increase in systemic iron [8]. This disparity suggests that the processes that regulate iron homeostasis could be overwhelmed by chronic inflammation; however, it is not yet known whether inflammation is specifically correlated to increases in brain iron levels.

Recently, research on the potential antioxidant capacities of carbon nanodots has emerged, and the mice for this project were received from a lab that looks at the use of carbon nanodots as a nanomaterial for biomedical research such as drug delivery, an anti-inflammatory, a bioimaging tool, and a potential antioxidant [9]. Not only have carbon nanodots been studied for their antioxidant and anti-inflammatory properties, but they have also become popular in research due to their excellent photoluminescence, biocompatibility, and stability [10]. Current research looking at the effects of carbon nanodots on inflammation has found that carbon nanodots reduce the expression of proinflammatory genes, act as an electron donor to support free radical scavenging, promote the phagocytic activity of macrophages, and can protect against oxidative damage from reactive oxygen species [9,10,11]. Specifically, one study looked at the effects of carbon nanodots on atherosclerosis, which corresponds with the physical state of the LDLr knockout mice used in the present study [10]. Each of these findings supports the hypothesis that carbon nanodots may be a potential treatment for chronic conditions in which an inflammatory state is present, which is why we also wanted to look at the differences in brain trace metals (iron and copper) based on varying levels of exposure to carbon nanodots in LDLr knockout mice.

The main goal of this study was to assess the antioxidant qualities of carbon nanodots in an in vivo model of oxidative stress. We utilized brain lipid peroxidation and elevated concentrations of brain iron as markers of oxidative stress. Our hypothesis was that treatment with carbon nanodots would attenuate the elevated brain lipid peroxidation and iron concentrations observed in the LDLr knockout mouse (our in vivo model of oxidative stress), whereas in mice without oxidative stress (C57BL/6J mice), exposure to carbon nanodots would have no neurological impact. Since the current therapies for the prevention and/or treatment of neurodegenerative diseases are scant, our study has significant health relevance.

## 2. Materials and Methods

### 2.1. Animals and Diet

Male C57BL6/J mice (5 weeks old; n = 20) were fed a standard chow diet, and LDL receptor (LDLr) knockout mice (6 weeks old; n = 20) were fed an atherogenic diet (TD:88137; Envigo Indianapolis, IN) for 16 weeks. Half of each mouse strain was injected with carbon nanodots (2.5 mg/kg bw) to assess their antioxidant abilities, and the other half of each mouse strain was injected with saline (0 mg/kg bw), for a total of four groups of animals, each with 10 mice. The number of animals required for this study was determined through preliminary studies and was based on a power analysis using estimated variance. All animal procedures were approved by the Animal Care and Use Committee at the University of North Carolina at Greensboro, Protocol #20-005.

### 2.2. Synthesis and Characterization of Carbon Nanodots

Carbon nanodots were synthesized based on our published microwave-assisted method [9,11,12]. The samples were diluted with 10.0 mL of DDI water and then dialyzed against 5.0 L of DDI water using a 500–1000 Da MWCO (Spectra Por Float-A-Lyzer G2, 10 mL, G235063; MilliporeSigma; Darmstadt, Germany). The sample was dialyzed over three days under a gentle stir with a new DDI water replacement every 24 h. The resultant clear, brown-orange aqueous solution was lyophilized (Labconco FreeZone Plus 12; Kansas City, MO, USA) to produce the dried carbon nanodots. The Cary^®^ Eclipse™ Fluorescence Spectrophotometer was used for characterizing the photoluminescence of carbon nanodots.

### 2.3. Elevated Plus Maze

Elevated plus maze experiments were employed to assess anxiety-like behavior. This test is used to assess unconditioned avoidance behavior in a novel environment (elevated, open spaces), which focuses on spontaneous behavior indicative of anxiety and requires no training. We used a maze constructed from plastic that is raised 60 cm from the ground and has a 10 × 10 cm square base with one arm coming from each side (making a plus). Two of the arms opposite each other are open (10 × 32.5 cm), while the other two arms opposite each other are “closed” (10 × 32.5 × 15 cm) such that it has 15 cm high walls on either side and at the ends of the arms. Mice were placed facing an open arm and allowed to explore for 5 min under video recording using HomeCageScan software. Variables measured via TopScan software were the ratio of open arm entries relative to total entries, latency to the first entry, and percent duration spent in closed versus open arms.

### 2.4. Brain Tissue Collection

Mice were humanely anesthetized with isoflurane, decapitated, and their brains removed and dissected into four brain regions: the midbrain, striatum, cortex, and “rest” of the brain. The brain tissues were stored at −80 °C until analysis in pre-labeled, RNAse free microcentrifuge tubes. The “rest” was not used for this study but saved in case we needed it for future research. We targeted these brain regions due to their known vulnerability to oxidative stress and their iron concentrations.

### 2.5. Protein Analysis

For total protein analysis, the brain tissue samples were first sonicated using the sonic dismembrator (Model 50, Fisherbrand, Fisher Scientific, Waltham, MA, USA) into a cold radioimmunoprecipitation-assay (RIPA) buffer until homogenized. Once sonicated, this homogenate was then used to determine protein concentration by using the Pierce Bicinchoninic Acid (BCA) Protein Assay Kit (Thermo Fisher Scientific Inc., Waltham, MA, USA).

### 2.6. Trace Metal Analysis

We followed our standard method as described in Totten et al., 2020 [13]. Briefly, 50 μL aliquots of the homogenate of brain tissue were digested into pre-labeled test tubes with 50 μL ultra-pure nitric acid for trace metal analysis for 4 days at room temperature (21 °C). After the first 12 h and every 24 h following, the sample tubes were opened to release trapped gas, closed, vortexed for 15 s, and placed back under the hood. At the end of the 4 days, 50 μL of the digested homogenate was then diluted based on the metal to be analyzed into pre-labeled test tubes with 2% nitric acid for trace metal analysis. Concentrations of copper and iron were determined using graphite furnace atomic absorption spectroscopy (GFAAS) (model AA240, Agilent Technologies Inc., Santa Clara, CA, USA). All samples were run in duplicates, as the Varian AA240 gives an accurate determination of metal concentrations. Bovine liver (NBS standard reference material, USDC, Washington, DC, USA) was digested in ultrapure nitric acid and used as an internal standard during analyses.

### 2.7. Thiobarbituric Acid Reactive Substances (TBARS) Assay

MDA levels, an indicator of lipid peroxidation, were measured in the midbrain, striatum, and cortex of C57BL6/J wild-type mice and LDLr knockout mice using the TBARS assay. Brain tissue samples were weighted and sonicated (three times per 15 s) with ice-cold 50 mM potassium phosphate buffer, pH 7.4, containing 2 mM EDTA and 0.1% Triton X-100, followed by centrifugation at 10,000× *g* for 5 min. The supernatant (50 μL) was collected from centrifuge tubes into new tubes and mixed with distilled water (150 μL), obtaining a final volume of 200 μL. To prepare the TBARS standard, a 10 mM 1,3,3-tetramethoxypropane stock standard was prepared by diluting 16 μL of TMOP in 10 mL of ethanol (storage at 4 °C). The working standard was prepared by dissolving 10 μL of stock into 4990 μL of distilled water. The malondialdehyde (MDA) standard curve was established by marking microcentrifuge tubes B (blank) and S1, S2, S3, S4, S5, and S6 (standard). To these tubes, 12.5, 25, 50, 100, 150, and 200 μL of the working standard were added, and the final volume was made up to 200 μL with distilled water. To all the samples and standards, 375 μL of 20% acetic acid, 375 μL of 0.8% TBA, 50 μL of 8.1% SDS, and 150 μL of deionized water were added in the respective order described. All tubes were then incubated at 95 °C for 60 min and cooled with tap water. The upper organic layer was aliquoted into a cuvette and measured at 532 nm. Results are expressed as micromoles of MDA per mg of tissue.

### 2.8. Statistical Analyses

Body weight, copper, iron, behavioral data, and lipid peroxidation data were analyzed using a general linear model with the main effects of carbon nanodot treatment and strain (C57BL/6J vs. LDLr Knockout). The significance level for these analyses was set at *p* < 0.05. A Tukey’s HSD test was used for post-hoc analyses. We utilized SPSS (version 28.0.0; IBM SPSS, Armonk, NY, USA) software to conduct all analyses.

## 3. Results

### 3.1. Body Weight

LDLr knockout mice gained significantly more weight than C57BL/6J mice (*p* < 0.001) (Table 1). While mice from both strains started the 16-week study around the same weight, the C57BL/6J mice gained an average of 8 g of body weight compared to the 22 g gained by the LDLr knockout mice. It is important to note that treatment with carbon nanodots did not affect body weight within each strain (Table 1).

### 3.2. Characterization of CNDs: UV-VIS

The Cary^®^ Eclipse™ Fluorescence Spectrophotometer was used to characterize carbon nanodots. The photoluminescent properties of carbon nanodots can be seen in their excitation wavelength around 360 nm and their emission peak around 460 nm (Figure 1), which is consistent with our previous reports [9,11].

### 3.3. Elevated Plus Maze

Exploratory behavior of the open arm in the elevated plus maze was similar between the wild-type mice and the wild-type mice receiving carbon nanodots treatment (Figure 2). LDLr Knockout mice spent significantly less time on the open arm (*p* = 0.025) compared to wild-type mice, and treatment with carbon nanodots in LDLr Knockout mice (2.5 mg/kg dose) normalized this behavior to match both wild-type groups (Figure 2).

### 3.4. Iron and Copper

In both the midbrain and striatum, iron concentrations were significantly elevated in the LDLr knockout (KO) mice compared to the C57BL6/J (WT) mice but were normalized with carbon nanodots treatment (*p* < 0.05) (Figure 3A,B). The striatum was the only brain region where an elevation in copper concentrations was observed in the LDLr knockout mice compared to the C57BL6/J mice (*p* < 0.05). This significant elevation in striatal copper concentrations was normalized with carbon nanodot treatment (Figure 4B). Carbon nanodot treatment had no effect on iron or copper levels in the C57 BL/6J (WT) mice’s brains (Figure 3 and Figure 4).

### 3.5. Malondialdehyde Levels Expressed as TBARS Content

TBARS content was significantly higher in the midbrain (*p* < 0.05) of LDLr knockout mice compared to those of C57BL6/J wild-type mice but was normalized by carbon nanodots treatment (Figure 5A). TBARS content was elevated in the cortex with a trend towards significance (Figure 5C, *p* < 0.10) compared to the C57BL6/J wild-type mice. Exposure to carbon nanodots did not have any significant lowering effects on TBARS content in these two brain regions of the LDLr knockout mice and had no effect on the levels of TBARS in the midbrain, striatum, and cortex in the brains of C57 BL/6J (WT) mice.

## 4. Discussion

To our knowledge, our study is the first to identify significant changes in brain regional iron levels and increased anxiety concomitant with lipid peroxidation in the LDLr knockout mouse model. Importantly, we show that treatment with carbon nanodots reduced lipid peroxidation, normalized iron levels, and improved anxiety-like behavior observed in an in vivo model of oxidative stress (LDLr knockout mouse). This marked attenuation of lipid peroxidation by treatment with carbon nanodots has implications for treating neurodegenerative disorders.

Carbon nanodots are an emerging nanomaterial with a diameter of less than 10 nm. These nanoparticles have unique properties such as excellent biocompatibility, low toxicity, good stability, excellent fluorescence properties, and easy surface modification [12,14,15]. These properties make it attract great attention and research in the field of biomedicine. Carbon nanodots have found applications in many fields, including bioimaging, biosensing, and photothermal therapy. The most prominent feature of the carbon nanodot is their ability to scavenge free radicals because they exhibit good electron-donating ability [12,14,15]. Reactive oxygen species, such as hydroxyl radicals and superoxide anions, are unstable molecules that tend to act as electron acceptors due to their unpaired outer electrons; thus, they tend to gain electrons from other molecules (e.g., lipids and nucleic acids), thereby disrupting their chemical structures. Previous studies have shown that carbon nanodots can scavenge superoxide and hydroxyl radicals in a dose-dependent manner [12,14,15]. Some studies have shown that the antioxidant activity of carbon nanodots may be attributed to the characteristics of some functional groups of this nanomaterial, such as primary amine and carboxyl groups [16,17]. In this study, it is reasonable to assume that the treatment with carbon nanodots reduced brain lipid peroxidation in the LDLr knockout mice due to their strong ROS scavenging abilities. Although the specific antioxidant mechanism(s) of carbon nanodots are not known, we hypothesized that altered iron and copper biology may be driving the oxidative stress in LDLr knockout mice’s brains. Alterations of these two metals are known to be pro-oxidants in the brain. Specifically, iron and copper engage in one-electron oxidation-reduction reactions. Ferrous iron (Fe^2+^) participates in Fenton reactions to produce highly reactive hydroxyl radicals (HO•), which cause oxidative damage to lipid membranes, proteins, and nucleic acids. The effect of brain iron dysregulation on lipid peroxidation is not well characterized, but it is important to understand metal-related neurodegeneration. Our finding that the LDLr knockout mice had significantly increased iron concentrations in the midbrain and striatum (Figure 3) is particularly noteworthy given the relationship of these brain regions to neurodegenerative diseases. The midbrain is of particular interest for the study of Parkinson’s disease, as this brain region is known to accumulate iron and alpha-synuclein as neurodegeneration progresses, including chronic inflammation [18,19]. Chronic inflammation has been shown to limit systemic iron availability, but brain region-specific iron toxicity has been seen without increases in systemic iron, which could lead to the assumption that inflammation could specifically affect brain iron homeostasis [8]. Interestingly, it has been reported that iron can accumulate in the striatum of multiple sclerosis patients due to inflammation of brain tissue and the attraction of iron-rich microglia [20]. During a state of neuroinflammation, microglia become activated to sequester extracellular iron to protect surrounding tissues, thereby increasing intracellular iron in this mode of defense [21]. Triggers for neuroinflammation can include proinflammatory cytokines, pathogens [21], or protein deposits such as β-amyloid [22] or alpha synuclein [23,24]. Systemically, it has been reported that dietary-induced obesity elicits a redistribution of iron (increased iron in adipose tissue and decreased iron in the liver), accompanied by upregulation of inflammatory cytokines and reduced ferromagnetic adipose tissue macrophages [25]. Since the LDLr knockout model displays a heightened state of inflammation, it is possible that the increase in iron that we see in our study may be related to this attraction and activation of microglia. Regardless of any potential mechanism driving this increased brain iron in the LDLr knockout, it is clear that increased oxidative stress is involved.

Oxidative stress occurs when reactive oxygen species are produced at a rate that far exceeds the body’s ability to clear them, especially under pathological conditions. It has been reported that the brain tissue from the LDLr knockout rodent model can produce a higher neuroinflammation response, generate excess reactive oxygen species, increase lipid peroxidation, and lead to mitochondrial dysfunction and antioxidant system damage [26,27]. The brain is particularly sensitive to oxidative damage, mainly due to the following features [28,29]. First, brain tissue has higher and more specific metabolic activity compared to other tissues. Second, the brain has extremely high oxygen consumption and is almost completely dependent upon oxidative phosphorylation for the generation of ATP. Finally, the brain is rich in iron, especially the midbrain and striatum, and has a higher concentration of lipids that are very prone to peroxidation [28,29]. All of these factors contribute to the hypersensitivity of brain tissue neurons to oxidative stress. Malondialdehyde (MDA) is a well-known biomarker for lipid peroxidation and oxidative damage. In this study, the TBARS method was used to detect lipid peroxidation as an indicator of oxidative stress in brain tissue. Results of this study showed that in wild-type mice, MDA levels in all tested brain tissues (cortex, striatum, and midbrain) did not change significantly between control and carbon nanodot treatment, indicating the safety of carbon nanodots in terms of oxidative stress. MDA levels in the midbrain and cortex were increased in untreated LDLr knockout mice compared with untreated C57BL/6 mice, suggesting that altered cholesterol metabolism in LDLr knockout mice produces oxidative damage in the brain. LDLr knockout mice treated with carbon nanodots overall showed a decrease compared to the untreated LDLr knockout animals, indicating the anti-oxidative stress properties of carbon nanodots.

Anxiety-like behavior was assessed by measuring the time mice spent in the open versus closed arms of an elevated plus maze, a well-characterized pre-clinical methodology [30]. We observed a significant reduction in open arm time in the LDLr knockout group, consistent with an anxiogenic phenotype [30]. The LDLr knockout mice are known to have impaired spatial cognition [31], but a previous examination of anxiety-like behavior using an elevated zero maze reported no anxiogenic development between LDLr knockout mice fed a high or low cholesterol diet compared to C57BL/6J mice [32]. Evidence of inflammatory damage and brain metals was not reported by Elder et al., and it is likely that a diet high in dietary cholesterol may not engage pro-inflammatory pathways to the same extent as the atherogenic, high-fat diet used in our study. We report elevated MDA in LDLr knockout mice indicative of oxidative damage, which paralleled the accumulation of iron and copper in dopaminergic striatal and midbrain regions, critical to locomotor activity. Other studies using the elevated plus maze showed treatments that reduced oxidative stress in a β-amyloid model of Alzheimer’s disease, increased open arm exploration in rats [33], attenuated lipid peroxidation in mice that overexpressed neuroglobin, and improved exploratory open arm time [34], implicating oxidative damage with this measure of anxiety. Moreover, iron overload was reported to increase anxiety-like behavior by reducing open arm time in rats [35], putatively through oxidative damage. Together, these studies show that an anxiogenic phenotype coincides with enhanced oxidative damage and dysregulated brain metal homeostasis. Our observation that treatment with carbon nanodots increased time spent in the open arm of the elevated plus maze and prevented oxidative damage in LDLr knockout mice suggests that the antioxidant function of carbon nanodots contributes to their anxiolytic effect. The fact that these nanoparticles also reduced iron and copper accumulation in LDLr knockout mice in specific brain regions that evaluate environmental stimuli and externalize behavioral control (midbrain and striatum) is strong evidence linking dysregulated metals and oxidative damage with anxiety-like behavior.

While our findings linking carbon nanodot treatment with the attenuation of brain lipid peroxidation are encouraging, our study does have an important limitation, i.e., the exclusive use of males. We have previously established that the impact of high-fat feeding on brain neurochemistry, trace element concentrations (iron and copper), and behavior shows a strong sex difference. When compared to mice fed a normal fat diet, males fed a high fat diet display abnormal brain trace element concentrations and behaviors, whereas females fed a high fat diet showed a greater impact on dopamine biology [13,36]. A study by Mansukhani et al. showed that female LDLr knockout mice fed the atherogenic diet had significantly higher cholesterol, LDL, and concomitant atherosclerosis when compared to male LDLr knockout mice [37]. These data would suggest that female LDLr knockout mice may be more vulnerable to brain lipid peroxidation than male mice, but the mechanism is likely different. Knowing that there are significant sex differences in the risk of neurodegenerative diseases, such as Alzheimer’s disease, the inclusion of both sexes in our future studies is critical and planned. These future studies will also include the assessment of endogenous antioxidant systems to elucidate the antioxidant mechanisms of carbon nanodots. The nuclear factor erythroid 2-related factor 2 (Nrf2) is a well-characterized regulator of endogenous antioxidants. Recently, some of these antioxidants were found to be upregulated in human endothelial cells treated with carbon nanodots [9]. It is likely that in the midbrain of LDLr knockout mice treated with carbon nanodots, the significant reduction in lipid peroxidation was due to an upregulation of these endogenous antioxidants.

## 5. Conclusions

Our data support the efficacy of treatment with carbon nanodots in combating the neurological effects of lipid peroxidation. We showed that in an in vivo model of lipid peroxidation (LDLr knockout mice), treatment with carbon nanodots effectively lowered indicators of lipid peroxidation (significantly elevated MDA [TBARS content] and iron concentrations) in the midbrain. Our finding is particularly salient given that the pathologies associated with Parkinson’s disease are centered in this brain region. In sum, our data lend support for pursuing treatment with carbon nanodots as a safe, effective therapy for attenuating neurodegenerative processes in vulnerable populations, especially where brain iron dysregulation and concomitant lipid peroxidation are involved.

## Figures and Tables

**Figure 1 antioxidants-12-01081-f001:**
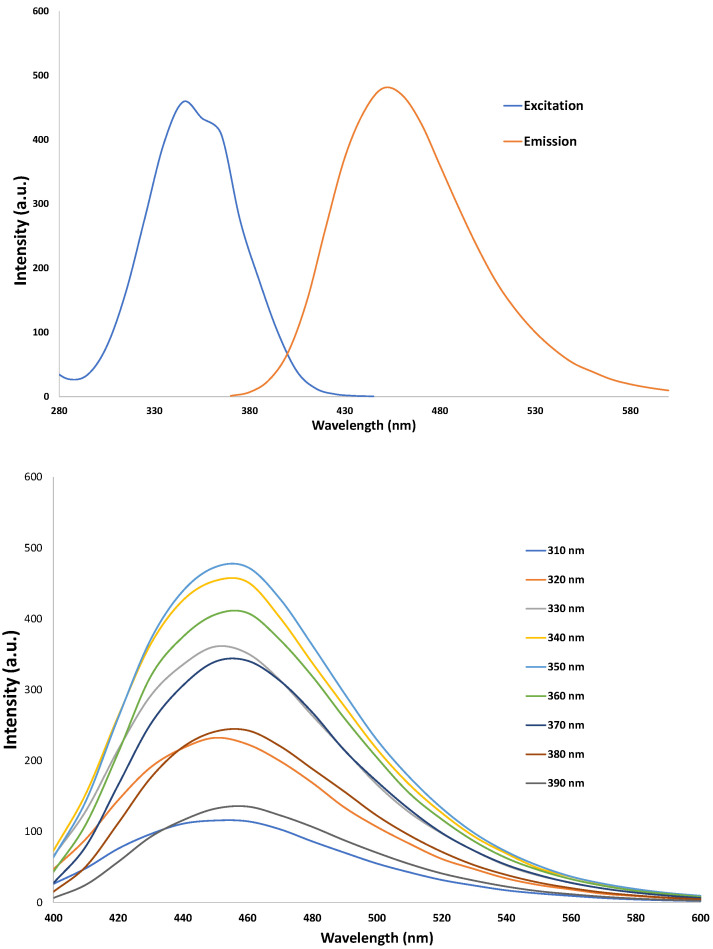
Characterization of carbon nanodots. (**Top**) panel: absorption spectrum. The emission peak is at 460 nm, with an excitation wavelength of 360 nm. (**Bottom**) panel: carbon nanodots show an emission peak of ≈460 nm with excitation wavelengths from 220 to 400 nm.

**Figure 2 antioxidants-12-01081-f002:**
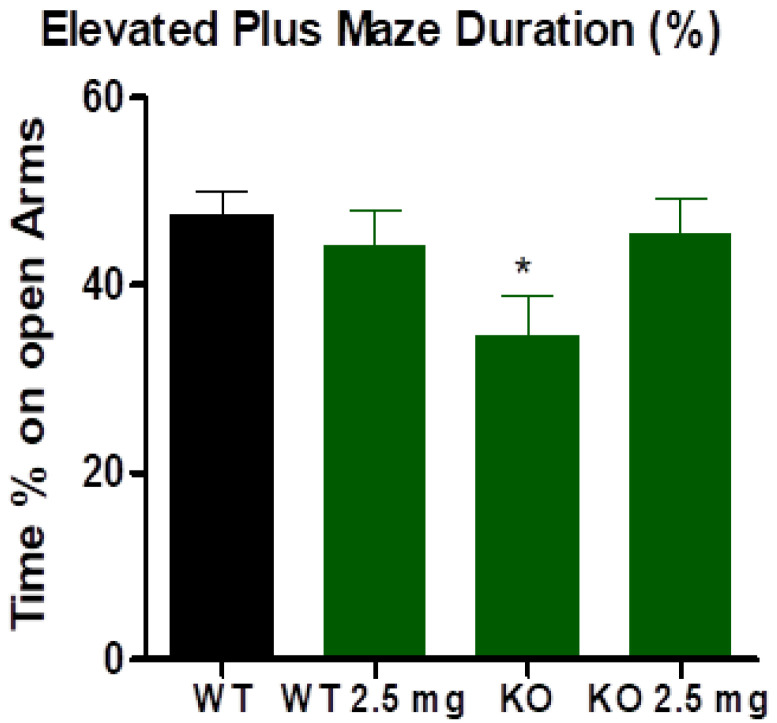
LDLr knockout (KO) mice have significantly higher anxiety than C57BL6/J (WT) mice (*p* = 0.025) as indicated by asterisk (*). In rodents, the less time spent in an open arm, the higher the anxiety.

**Figure 3 antioxidants-12-01081-f003:**
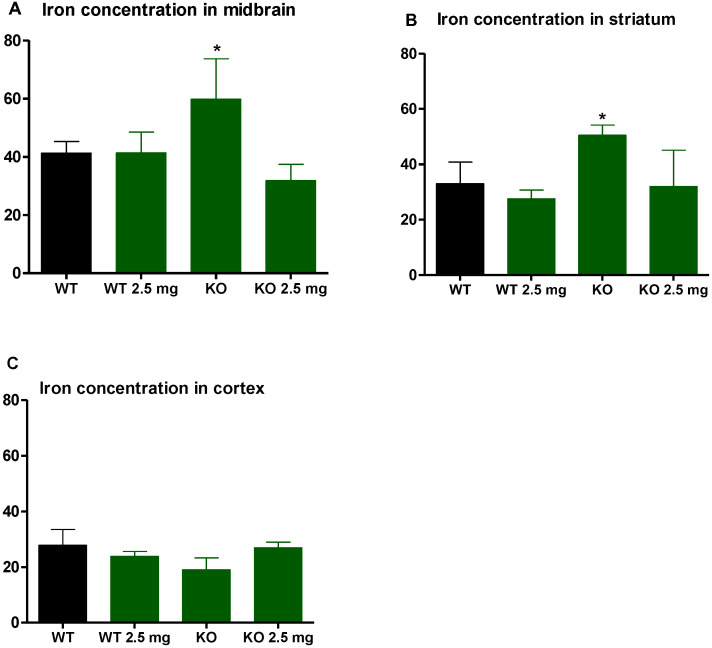
Brain iron concentrations are expressed as ug Fe/mg protein. Data are mean ± standard error in the midbrain (**A**), striatum (**B**), and cortex (**C**) of C57BL6/J (WT) mice and LDLr knockout (KO) mice treated with 0 or 2.5 mg carbon nanodot for 16 weeks. LDLr knockout mice showed significantly elevated iron in both the midbrain and striatum, as indicated by the asterisk (*) (*p* < 0.05). This significant elevation was normalized with the carbon nanodot treatment.

**Figure 4 antioxidants-12-01081-f004:**
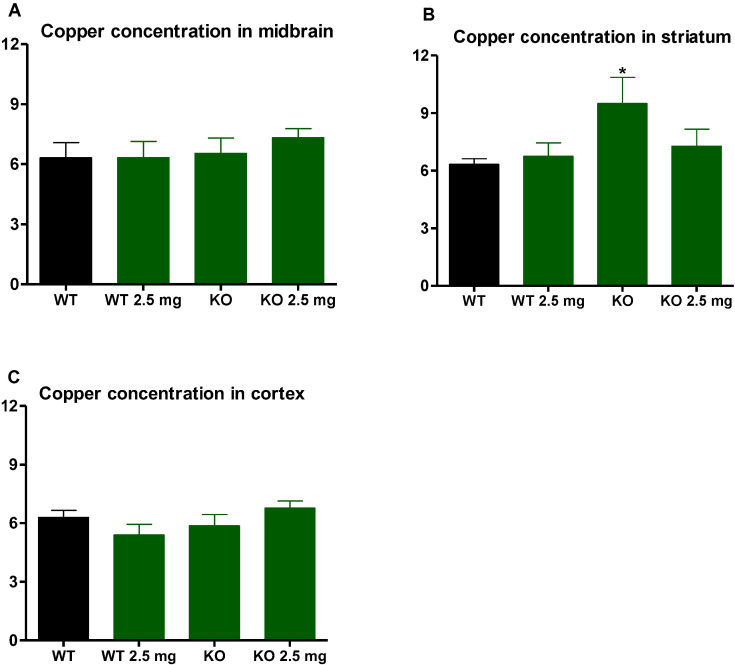
Brain copper concentrations expressed as ug Cu/mg protein. Data are mean ± standard error in the midbrain (**A**), striatum (**B**), and cortex (**C**) of C57BL6/J (WT) mice and LDLr knockout (KO) mice treated with 0 or 2.5 mg carbon nanodots for 16 weeks. LDLr knockout mice showed significantly elevated copper in the striatum, as indicated by the asterisk (*) (*p* < 0.05). This significant elevation was normalized with the carbon nanodot treatment.

**Figure 5 antioxidants-12-01081-f005:**
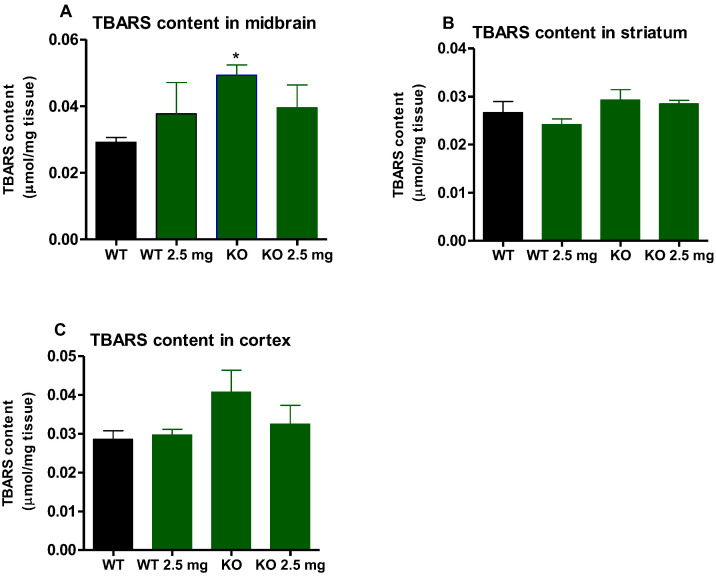
Malonaldehyde levels are expressed as TBARS mean ± standard error in the midbrain (**A**), striatum (**B**), and cortex (**C**) of C57BL6/J (WT) mice and LDLr knockout (KO) mice treated with 0 or 2.5 mg CND for 16 weeks. LDLr knockout mice showed significantly elevated TBARS in the midbrain, as indicated by the asterisk (*) (*p* < 0.05). This significant elevation was normalized with the carbon nanodot treatment.

**Table 1 antioxidants-12-01081-t001:** Body weight and weight gain data.

Treatment Group	Initial Body Weight (day 1, g)	After Treatment Body Weight (16 Weeks, g)
C57BL/6J Saline Control	23.08 ± 0.44 ^a^	33.92 ± 1.03 ^a^
C57BL/6J CND 2.5 mg/kg	23.15 ± 0.63 ^a^	30.71 ± 0.81 ^a^
LDLrKO CND 0 mg/kg	21.80 ± 0.32 ^a^	42.79 ± 1.05 ^b^
LDLrKO CND 2.5 mg/kg	22.08 ± 0.33 ^a^	44.50 ± 1.27 ^b^

Data are expressed as mean ± standard error. Statistical differences are indicated by superscript letters. CND: Carbon nanodots.

## Data Availability

Data associated with this manuscript may be requested from the corresponding author. Keith Erikson: kmerikso@uncg.edu; Zhenquan Jia: z_jia@uncg.edu.

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
