# Peer review of "Carbon Nanodots Attenuate Lipid Peroxidation in the LDL Receptor Knockout Mouse Brain"

_antioxidants, 2023, doi:10.3390/antiox12051081_

Round 1
Reviewer 1 Report
Remarks to the Authors
The subject of the paper antioxidants-2349096 entitled “Carbon nanodots attenuate lipid peroxidation in the LDL receptor knock-out mouse brain” seems to be interesting. However, very numerous questions need attention and should be improved.
The main questions that should be addressed are mentioned below.
Point-by-point remarks to the Authors
1) Extensive English language correction of the article is necessary. Some parts of the manuscript were written in non-scientific language. For example:
- Lines 11-12: “Low-density lipoprotein receptor are models for studying altered cholesterol metabolism and oxidative stress onset in the brain.” It is obvious that the Authors had animals in mind; however, the text implies that the LDL receptors, not the LDL knockout mouse, are the experimental model.
- Lines 15-17: “Brains were removed and dissected into cortex, midbrain and striatum and analyzed for lipid peroxidation, iron and copper.” Wording like “analyzed for lipid peroxidation, iron and copper” should not be used in scientific work.
- Lines 17-19: We measured lipid peroxidation in the mouse brain tissues using the Thiobarbituric Acid Reactive Substances Assay and Graphite Furnace Atomic Absorption Spectroscopy to measure brain regional iron and copper.”
- Lines 22-24: “CND treatment in the LDLr knockout mice attenuated both the rise in iron and lipid peroxidation but they had no negative effect in the C57BL/6J mice indicating the anti-oxidative stress properties of CNDs.”
- Lines: 44-46: “These metals play important roles in cellular responses, signaling, regulation, and overall cell function and metabolism, making them necessary for normal function of the brain [5].”
- Lines 48-50: “When these metals become altered or disrupted there can be detrimental effects on the brain, such as in cases of neurodegenerative diseases [6].”
- Lines 56-59: “This suggests that the processes that regulate iron homeostasis could be overwhelmed by a condition that involves chronic inflammation, and it is not yet known whether inflammation is specifically correlated to increases in brain iron levels.”
2) Lines 12-13; “Carbon nanodots (CNDs) are a new class of carbon nanomaterials
below 10nm and have antioxidant properties” It is necessary to describe precisely what size of nanoparticles has been given (diameter).
3) lines 19-20: “Elevations of both of these metals in the brain are associated with oxidative stress”. This sentence is redundant in the abstract and should be removed.
4) line 20: “Iron levels” – the word “levels” is inappropriate to describe concentration.
5) Lines 22-24: “CND treatment in the LDLr knockout mice attenuated both the rise in iron and lipid peroxidation but they had no negative effect in the C57BL/6J mice indicating the anti-oxidative stress properties of CNDs.” The Authors are asked to reanalyze the sense of this sentence.
6) lines 25-26: The treatment with CND "eradicated" whether protected from such behaviors?
7) line 32: “Atherosclerosis is associated with ….”. It is a too general statement and may be unclear for a reader. How is the association?
8) The aim of the study has not been precisely described.
9) line 108: what anesthetic was used?
10) line 113: “iron biology”?
11) lines 119-121: how was the precision?
12) Very numerous typographical errors are present in the manuscript. For example: 50uL, 150 ul, 50ml, 16 mL
12) Some abbreviations used in the manuscript are unexplained.
13) line 163: “As anticipated…” In the Results section, the results of the study should be objectively presented without providing any opinion of the Authors.
14) The titles Table 1 and Figures 2-5 are very untypical. The titles should only describe what kind of data is presented in the table or figure (without a description of the results). All additional data in Table 1 should be provided below the table (not in the title of the table).
15) subsection 3.1. Body Weight; The body weight gain (not only the final body weight) should be provided. There is no data on the initial body weight. It is very important because, at the beginning of the study, the C57 BL6/J mice and LDL-receptor knockout mice were not the same age.
16) Figure 1 is poorly legible
17) Lines 186-190; 3.3. Iron and Copper
“In both the midbrain and striatum, iron increased in the LDLr knockout (KO) mice, and decreased with CND treatment (p=0.035) (Figure 3), but for copper levels, only the striatum was impacted (Figure 4). CND treatment had no effect on iron or copper levels in the C57 BL/6J (WT) mice brains.” Such a description is unacceptable in scientific work.
18) lines 206-208; This sentence is a repetition of methodology.
19) lines 208-212; “As shown in Figure 5, compared with C57BL6/J wild-type mice, the content
of TBARs in the midbrain of LDLr knockout mice was significantly increased (p<0.05), whereas the content of TBARs in the midbrain of LDLr knockout mice treated with CND was decreased. Changes in TBARs across different groups in the brain cortex tissues showed the same overall trend as in the midbrain, but not in the striatum.” The word “higher” is more appropriate to be used instead of “increased” to express that the concentration of TBARs in the midbrain of LDLr knockout mice was higher than in C57BL/6J mice. Moreover, it is not proper the statement that “the content of TBARs in the midbrain of LDLr knockout mice treated with CND was decreased”. The treatment with CND protected from the LDLr knockout-related increase in the concentration of TBARs in the midbrain/normalized the concentration.
The authors are asked to consider whether the claim “Changes in TBARs across different groups in the brain cortex tissues showed the same overall trend as in the midbrain, but not in the striatum” is justified by the results.
20) The discussion section needs to be improved. There are numerous repetitions of the results (or data provided in the Materials and Methods section) and nonscientific expressions (for example “the increase in iron”, “Evidence of inflammatory damage and brain metals were not reported…”, “…is strong evidence linking dysregulated metals and oxidative damage with anxiety-like behavior”).
21) Lines 222-223: “Second, the brain has extremely high oxygen consumption and is almost completely oxidative phosphorylation.” What do the Authors mean?
22) The Authors did not indicate and discuss any limitations of the study.
22) Conclusions should be clearly formulated.
23) List of references should be prepared according to the Instruction for Authors. All inaccuracies in the preparation of the list of references should be corrected.
24) lines 150-152: “The MDA amounts for each sample were calculated using the absorbance value and the equation generated from the standard curve”. This description will be unclear for a reader.
25) line 153; “in each sample” – it is redundant.
Extensive English language correction of the article is necessary. Some parts of the manuscript were written in non-scientific language. For example:
- Lines 11-12: “Low-density lipoprotein receptor are models for studying altered cholesterol metabolism and oxidative stress onset in the brain.” It is obvious that the Authors had animals in mind; however, the text implies that the LDL receptors, not the LDL knockout mouse, are the experimental model.
- Lines 15-17: “Brains were removed and dissected into cortex, midbrain and striatum and analyzed for lipid peroxidation, iron and copper.” Wording like “analyzed for lipid peroxidation, iron and copper” should not be used in scientific work.
- Lines 17-19: We measured lipid peroxidation in the mouse brain tissues using the Thiobarbituric Acid Reactive Substances Assay and Graphite Furnace Atomic Absorption Spectroscopy to measure brain regional iron and copper.”
- Lines 22-24: “CND treatment in the LDLr knockout mice attenuated both the rise in iron and lipid peroxidation but they had no negative effect in the C57BL/6J mice indicating the anti-oxidative stress properties of CNDs.”
- Lines: 44-46: “These metals play important roles in cellular responses, signaling, regulation, and overall cell function and metabolism, making them necessary for normal function of the brain [5].”
- Lines 48-50: “When these metals become altered or disrupted there can be detrimental effects on the brain, such as in cases of neurodegenerative diseases [6].”
- Lines 56-59: “This suggests that the processes that regulate iron homeostasis could be overwhelmed by a condition that involves chronic inflammation, and it is not yet known whether inflammation is specifically correlated to increases in brain iron levels.”
Author Response
We thank the Reviewer for their detailed review of our paper and we feel that we addressed all of their comments. We have attached a Pdf document with each of the Reviewer's comments/critiques and our response is in red font.

Reviewer 2 Report
Overall ok,
Figure 3A Is it normalized?
Figure 3C K.O. level is lower in the model; is there a relation to Figure 2? Anxiety iron etc
Figure 4A fine
Figure 4C needs further explanation.
Figure 5A needs further explanation of the possible anomaly
Line 276.
Basic grammatical errors involving uncountable nouns
Author Response
We thank the Reviewer for their careful review of our paper. We have addressed each of their comments in the attached pdf document. Our responses are in red font.

Round 2
Reviewer 1 Report
The revised paper antioxidants-2349096 entitled “Carbon nanodots attenuate lipid peroxidation in the LDL receptor knock-out mouse brain” has been markedly improved. The Authors responded to numerous remarks provided in the review. However, some questions still need attention and should be addressed.
The questions that should be addressed:
- The description “carbon nanodots” should be used throughout the manuscript, whereas in numerous places the Authors used “carbon nanodot”
- there are still numerous typographical errors in the manuscript (for example TBARS, TBAR, and TBARs)
- line 264: •OH, not OH• should be used for hydroxyl radical
- lines 77-78: “elevated brain iron” (elevated concentration of iron in the brain
- English editing still seems to be necessary (for example lines 86-90)
- lines 222-224: “TBAR content..” TBAR?, content whether concentration? Moreover, there is a lack of consistency regarding the concentration of TBARS in the cortex of the brain. The Authors have stated that “TBAR content was higher in the midbrain (p<0.05) and cortex (p<0.10) of LDLr knockout mice compared to those of C57BL6/J wild-type mice…(Figure 5).” However, this difference has not been marked in Figure 5C.
- In the conclusions, the Authors repeated the results.
-
Author Response
We thank Reviewer 1 for their second review of our manuscript. Please find our responses to their comments in the attached file (our responses are in red font).
